# Response of *Cucumis sativus* to Neighbors in a Species-Specific Manner

**DOI:** 10.3390/plants12010139

**Published:** 2022-12-27

**Authors:** Xiu Zhang, Jingfan Yan, Fengzhi Wu

**Affiliations:** 1Department of Horticulture, Northeast Agricultural University, Harbin 150030, China; 2Key Laboratory of Cold Area Vegetable Biology, Northeast Agricultural University, Harbin 150030, China

**Keywords:** competition, behavioral strategies, growth, allocation, plant interaction

## Abstract

Plants exhibit various behaviors of growth and allocation that play an important role in plant performance and social interaction as they grow together. However, it is unclear how *Cucumis sativus* plants respond to different neighbors. Here, we performed 5 neighbor combinations with *C. sativus* as the focal species. The selected materials of *C. sativus* responded to neighbors and exhibited different behavior strategies in a species-specific manner. All competition treatments reduced the growth of *C. sativus* seedlings to a certain extent, but only the *Eruca sativa* neighbor treatment reached a significant level in total root length and shoot biomass. Compared with growing under solitary conditions, focal plants avoided, tended to and did not change their allocation to their nearby plants. The larger the biomass of their neighbors, the stronger the inhibition of the focal plants. In addition, no significant correlations between growth and allocation variables were found, suggesting that growth and allocation are two important aspects of *C. sativus* behavioral strategies. Our findings provide reference and support for agricultural production of *C. sativus*, but further research and practice are still needed.

## 1. Introduction

The close association of neighboring plants, coupled with the strong similarity in resource demands, leads to competition for limited ecological resources, such as light, water, and nutrients [1]. Competition can reduce individual growth, reproduction, and survival, which in turn affects species evolution and community structure [2,3]. Plants are very intelligent in that they have the ability to predict possible competitive behaviors and trade off cost and benefit, which can maximize fitness of individuals or communities [4]. Better understanding of plant communication and coexistence can be achieved by understanding the behavioral strategies of plants when encountering competition.

There is adequate evidence that plants can perceive and respond behaviorally to their neighbors [5,6,7]. According to game theory, reduced nutrient availability due to competitive encounters results in excessive root proliferation, which is coined as the tragedy of the commons [8]. This idea has been supported in *Phaseolus varigaris* and *Triticum aestivum* [9,10]. However, not all reports indicated such a phenomenon. Semchenko et al. [11] found that *Avena sativa* plants that underwent root competition performed as well as those that did not. Based on this situation, the ideal free distribution, an alternative growth model for the tragedy of the commons, was proposed and verified in *Brassica rapa*. This model assumes that plant root growth responds only to nutrient availability, and the presence of neighbors may lead to depletion of nutrient resources [12]. Furthermore, there are instances showing that *Abutilon theophrasti*, *Rumex palustris*, *Agrostis stolonifera*, *Poa cita*, *Poa colensoi*, and *Helianthus annuus* roots were reduced in the presence of neighbors [5,13,14,15].

Plants can display avoidance, confrontation, and tolerance to their neighbors at different temporal and spatial scales. Studies have shown that some plants, such as *Arabidopsis thaliana*, *Hieracium pilosella*, and *Capsicuum annum*, may reorient leaf and root growth to avoid competition [16,17,18]. Roots of *Glechoma hederacea*, *Zea mays*, and *Oryza sativa* also exhibited root avoidance responses in the presence of conspecific or heterogeneic neighbors [11,19,20,21]. However, avoidance behavior may be adaptive in situations where competition is low as plants have to confront competition in some cases. When competitive confrontation takes, plants incur costs to promote resource utilization, which indirectly increases the negative impact of competition. Additionally, individuals can switch to tolerance strategies under long-term competitive challenges [4]. Furthermore, the foraging scale–precision trade-off assumption, made by Campbell et al. [22], indicated that the subdominant species may persist by foraging more precisely in small resource patches missed by the dominant species. Bartelheimer et al. [23] indicated that *Corynephorus canescens*, *Festuca psammophila*, *Hieracium pilosella*, *Hypochoeris radicata*, and *Conyza canadensis* aggregated roots towards interspecific neighbors. *Z. mays*, *Glycine max*, and *O. sativa* also showed root aggregation towards their neighbor in certain cases [19,20].

In addition to the competitive actions mentioned above, allelopathy and facilitation also interfere with behavioral outcomes during competition. Common examples of allelopathic disturbances are that allelopathic and invasive plants release chemicals to inhibit the growth of weeds and native plants [21,24,25,26]. Meanwhile, a recent study also demonstrated that allelochemicals mediated root placement patterns in interspecific interactions [27]. For facilitation, the nitrogen fixation and phosphorus mobilization of legumes modifies the root morphogenesis of other plants, thereby achieving interspecific facilitation in traditional intercropping systems [28,29,30]. Therefore, elucidating the growth and allocation strategies of species can improve cultivation and management and, thus, contribute to economic benefits in agriculture [31].

Continuous monoculture adopted in the cultivation of *Cucumis sativus*, an important horticultural crop, has caused problems such as yield reduction and quality deterioration [32,33]. Intercropping has proved to be a sustainable way to solve these problems [34]. The morphological plasticity is very important for the utilization of limited resources in the presence of neighbors in the intercropping systems. However, it is unclear how *C. sativus* plants respond to different neighbors. In this study, we tested behavioral outcomes in the growth and allocation responses to 5 different species belonging to 5 families in *C. sativus* to understand (1) the behavioral strategies of *C. sativus* plants in response to different neighbors and (2) correlations between 8 response variables, which was expected to provide a theoretical basis for species selection in intercropping system.

## 2. Results

Plastic responses of focal species were found in neighbor combinations (Figure 1, Figure 2 and Figure 3). In terms of response variables of growth, *C. sativus* showed decreases in the competition treatments compared with the solitary treatment to a certain extent (Figure 2). Among them, the *Eruca sativa* neighbor treatment significantly reduced the total root length (Figure 2B) and shoot biomass (Figure 2C) of *C. sativus*, but not the plant height (Figure 2A) or root biomass (Figure 2D). However, all growth indicators of *C. sativus* were not significantly affected by *C. sativus*, *Solanum lycopersicum*, *Chrysanthemum coronarium*, or *G. max* neighbor treatments (Figure 2). For response variables of allocation, on the other hand, *C. sativus* appeared to increase, decrease, or have no change in the neighbor treatments compared with the no neighbor treatment (Figure 3). Our results showed that *E. sativa*, *C. coronarium*, and *G. max* neighbor treatments resulted in horizontal asymmetry in roots but not in the *C. sativus* and *S. lycopersicum* neighbor treatments (Figure 3B–D). No significant effect was observed in the root:shoot ratio between all neighbor treatments (Figure 3A). The analysis of relative interaction index (RII) showed that the growth of *C. sativus* was inhibited by all competition treatments (Figure 4), and the inhibition of focal plants was stronger if the biomass of neighbors was larger. (Figure 5).

Correlation analysis between 8 response variables demonstrated a significant positive correlation between root biomass and total root length (Figure 6). Meanwhile, we found that there was significant agreement in the horizontal asymmetry results obtained by different variables (root width, total root length, and root biomass) (Figure 6). However, no significant correlations were found between variables of growth and allocation (Figure 6).

## 3. Discussion

Ecologists commonly agree that the plasticity of growth and allocation is significant for resource foraging encountering competition. Plants adopt different behavioral strategies in the presence of neighbor plants. Numerous studies have shown that plant traits, such as biomass, total root length, horizontal asymmetry etc., may change when encountering intraspecific or interspecific competition [8,20,35,36,37,38,39]. In our study, *C. sativus* seedlings exhibited a large number of behavioral responses to different neighbors, and the effects of neighbors were species-specific, suggesting that *C. sativus* may have the ability to recognize neighbor identities. Previous studies have also demonstrated that plants can distinguish neighbor identities, although they investigated traits between kin/non-kin [20,21] or conspecific/heterospecific plants [19,30] rather than different species. In this study, the growth of *C. sativus* was reduced in the presence of *E. sativa*, whereas the horizontal allocation of the *C. sativus* root system was altered in the *E. sativa*, *C. coronarium*, and *G. max* treatments. We discuss possible reasons for these behaviors below.

What factors are involved in the growth response of *C. sativus* plants to neighbors? We did not find the outcome that root over-proliferation occurs in the presence of neighbors, indicating that our experimental results do not support the theory of tragedy of the commons [8]. Furthermore, the results for the size of neighbors showed that the larger the biomass of neighbors, the smaller the biomass of focal plants, which means that it is clear that our results do not match the ideal free distribution model either [12]. It has also been suggested that root growth is influenced by nutrients and neighbors, and that plant performance is reduced due to competition [5]. Similar results were found in our current study where the plant height, total root length, and shoot and root biomass of *C. sativus* decreased to varying degrees in all 5 neighbor treatments. Therefore, we speculated that the growth response of *C. sativus* to changing neighbors may be related to reduced nutrient availability due to competitive encounters and root interactions.

On account of the differences in the characteristics of species, the focal plants showed unequal biomass in the different neighbor context [40]. Similar results were found in our study. Previous studies have shown that root exudates play a decisive role in plant interactions [41,42]. It has been reported that flavonoids secreted by leguminous plants contribute to nitrogen fixation, and carboxylates to phosphorus acquisition [43,44]. Furthermore, allelopathic autotoxicity occurs due to some chemicals secreted by *C. sativus* itself, such as cinnamic acid [45,46]. In our study, no significant differences were found in all growth response variables in the *G. max* neighbor or intraspecific competition treatments compared with solitary growing conditions. The reason may be that the kinds of chemical substances secreted by the selected materials are different from those of previous studies or that the amount secreted is not enough to cause allelopathy.

On the other hand, no growth response does not equal no response to neighbors. There is substantial evidence that some plants can recognize neighbors at the root level and, thus, alter root allocation [20,38,39]. Similar results were found in our study where the horizontal distribution of *C. sativus* roots was altered in the *E. sativa*, *C. coronarium*, or *G. max* neighbor treatments but not in the *C. sativus* and *S. lycopersicum* treatments. Among them, *C. sativus* grew away from *E. sativa* and toward *C. coronarium* and *G. max*. Plasticity of allocation in response to neighbors has been shown to have important effects on ecological processes [21,47]. Root segregation has been considered a common behavioral strategy that plants adopt for coexistence [18,48]. In this study, *C. sativus* seedlings exhibited similar behavioral strategies only in the competition of *E. sativa*. In addition, roots of *C. sativus* clustered towards *C. coronarium* and *G. max*, which may indicate that plants had to face competition [4]. Furthermore, previous studies have illustrated that legumes can change root morphology by mobilizing nutrients such as nitrogen fixation and phosphorus foraging [1,30]. The reason why *C. sativus* roots grew towards *G. max* may be the increase in nutrient availability by *G. max*. Combined with the above mentioned, no significant reduction in *C. sativus* growth was observed in the *G. max* treatment even though the root biomass of *G. max* was large. Therefore, we speculate that *G. max*, even Fabaceae plants, may be the appropriate plants for intercropping with *C. sativus*. Further research is needed to determine the specific mechanism.

Meanwhile, no significant correlation was found between growth and allocation responses, which suggested that growth and allocation are two aspects of plant behavioral strategies. Understanding plant behavioral strategies requires consideration of different aspects simultaneously. These results are consistent with the findings of Belter and Cahill [35]. However, in this study, the experimental results have certain limitations due to the small quantity of species and the short period of the experiment.

## 4. Materials and Methods

### 4.1. Plant Materials and Rhizoboxes

We chose five species belonging to five families: Cucurbitaceae (*C. sativus*); Solanaceae (*S. lycopersicum*); Brassicaceae (*E. sativa*); Asteraceae (*C. coronarium*); Fabaceae (*G. max*). All else being equal, plant roots proliferate preferentially in high nutrient areas [8]. We constructed heterogeneous soil treatments in order to determine the plasticity of selected cultivars of *C. sativus* (Figure 7). All seeds were supplied by the Department of Horticulture, Northeast Agricultural University.

Rhizoboxes were constructed to visualize roots [35,49]. The surrounds of the rhizoboxes were made of polyvinyl chloride (PVC), and one side was a detachable transparent plexiglass plate. The rhizoboxes were 30 cm long, 10 cm wide, and 40 cm deep (Figure 8A).

### 4.2. Experimental Design

#### 4.2.1. Plasticity Test of *C. sativus*

We tested the plasticity of *C. sativus* in local nutrient treatments. As shown in Figure 7, we designed 3 heterogeneous nutrient treatments: 0P (no nutrient-rich patch), 1P (with 1 nutrient-rich patch), and 2P (with 2 nutrient-rich patches). Each treatment was replicated 8 times. *C. sativus* was transplanted near the plexiglass at 15 cm from the edge of the rhizobox and 4 cm away from the two nutrient-rich zones (4 cm long) (Figure 7). CO(NH_2_)_2_, Ca(H_2_PO_4_)H_2_O, and K_2_SO_4_ fertilizers were applied resulting in 100 mg kg^−1^ available nitrogen (N), 100 mg kg^−1^ available phosphorus (P), and 100 mg kg^−1^ available potassium (K) in the nutrient-rich patches. After 30 d of growth, we evaluated shoot biomass, root biomass, proportion of root biomass, and root length in nutrient-rich zones to determine the plasticity of *C. sativus*. The results showed that the selected materials can be further studied (Appendix A); thus, *C. sativus* was used as the focal species for all five species as neighbors.

#### 4.2.2. Plasticity Test of *C. sativus* to Different Neighbors

The experiment consisted of focal species × neighbor (5) + focal species × no neighbor combinations; 6 treatments in total. The descriptions of each treatment is shown in Table 1. A focal plant and a neighboring plant were planted near the plexiglass at 10 and 20 cm from the edge of the rhizobox (Competition) (Figure 8B). A focal plant was planted near the plexiglass in the center of the rhizobox as a control for no neighbor (Solitary) (Figure 8C). Each treatment was replicated 8 times. Rhizoboxes were filled with a homogeneous soil composition of 3:1 sand:topsoil mix that contained (kg^−1^) 1.25 g organic carbon (C), 8.33 mg NH_4_^+^-N, 14.57 mg NO_3_^−^-N, 22.77 mg available P (AP), 23.57 mg available K (AK), 0.27 mS cm^−1^ electrolytic conductivity (EC) and had a pH of 7.16. During the entire experiment, all plants were carefully watered every 3 days to keep the substrate surface moist, but so that no liquid flowed out from the bottom of the rhizoboxes.

### 4.3. Growth, Visualization and Harvest

The experiment was carried out in a plastic-roofed greenhouse of Northeast Agricultural University. All plants were precultured by raising them from seeds. They were raised on a 3:1 sand:topsoil mix lasting approximately 15 days. Plants were selected for similar size within species and, as far as possible, also among species before roots were pruned to 3 cm length, which increased initial similarity between roots [29]. Seedlings were then transplanted into the rhizoboxes (Figure 8). The rhizoboxes were placed at an angle of 45°, with the clear side facing down, enhancing our ability to visualize root growth [50,51,52]. The clear plexiglass was covered with a black plastic sheet to prevent root exposure to light. After 30 d of growth, the roots were photographed for visualization before harvest [14]. We constructed 8 response variables for focal species to describe plant growth and allocation responses. The details of each response variable are shown in Table 2. In order to better understand the plant interaction, neighbor plants were harvested and the total biomass was counted. Meanwhile, the RII was calculated according to the following formula: RII = (Bw − Bo)/(Bw + Bo) [53], where Bw is the value of total biomass for the focal plant when grown with the neighbor plant, and Bo is the value of total biomass for the focal plant when grown alone. If 0 < RII ≤ 1, the neighbor plant promotes growth of the focal plant. If −1 ≤ RII < 0, the neighbor plant inhibits growth of the focal plant. If RII = 0, the neighbor plant neither promotes nor inhibits growth of the focal plant.

### 4.4. Statistical Analysis

One-sample *t*-tests were performed to analyze the effects of neighbors on the focal plants using IBM SPSS software (version 25) for each of the 8 response variables. Correlation analysis between variables was performed using the package “corrplot” in the R software (version 4.1.3). The one-way ANOVA and Tukey’s test were performed to test the differences in shoot and root biomass between neighbors, RII between competition treatments, and root plasticity response variables between heterogeneous nutrient treatments using IBM SPSS software (version 25).

## 5. Conclusions

*C. sativus* plants exhibited different behavioral strategies, including growth and allocation responses, when encountering different competitors. We found no evidence that *C. sativus* seedlings grow in accordance with the tragedy of the commons or the ideal free distribution theory. Furthermore, *C. sativus* seedlings might avoid or face encounters, and behavioral changes should be understood in conjunction with species characteristics. The larger the biomass of neighbors, the stronger the competition, the smaller the biomass of focal plants. Here, we did not find significant correlations in the growth and allocation responses of *C. sativus* plants to their neighbors. Our study provides a theoretical basis for the agricultural production of *C. sativus*, but further investigations are still needed.

## Figures and Tables

**Figure 1 plants-12-00139-f001:**
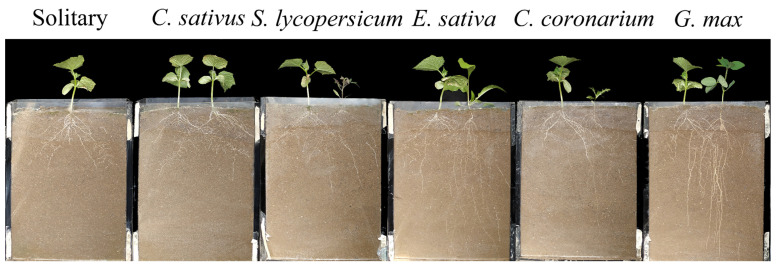
Phenotypic plasticity responses of roots to different neighbor combinations.

**Figure 2 plants-12-00139-f002:**
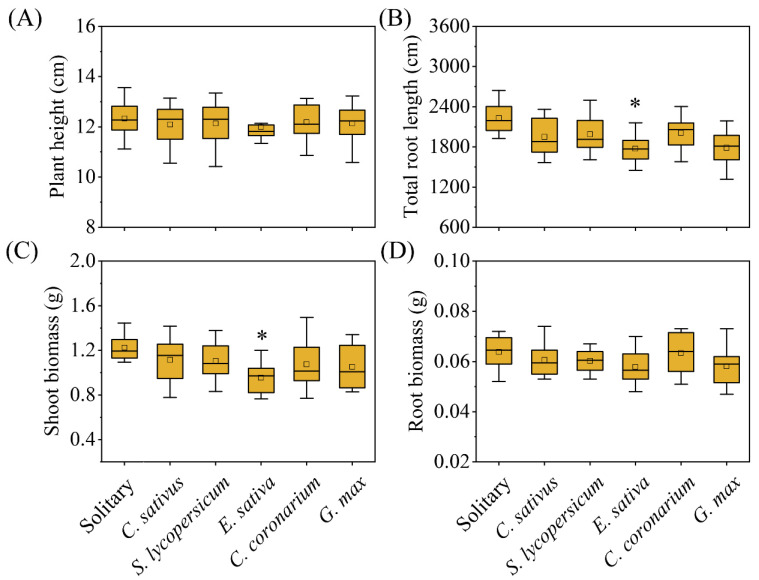
The growth responses of *C. sativus* to different competition treatments. * indicates a significant difference in (**A**) plant height, (**B**) total root length, (**C**) shoot biomass, and (**D**) root biomass between competition and solitary treatment (*p* < 0.05).

**Figure 3 plants-12-00139-f003:**
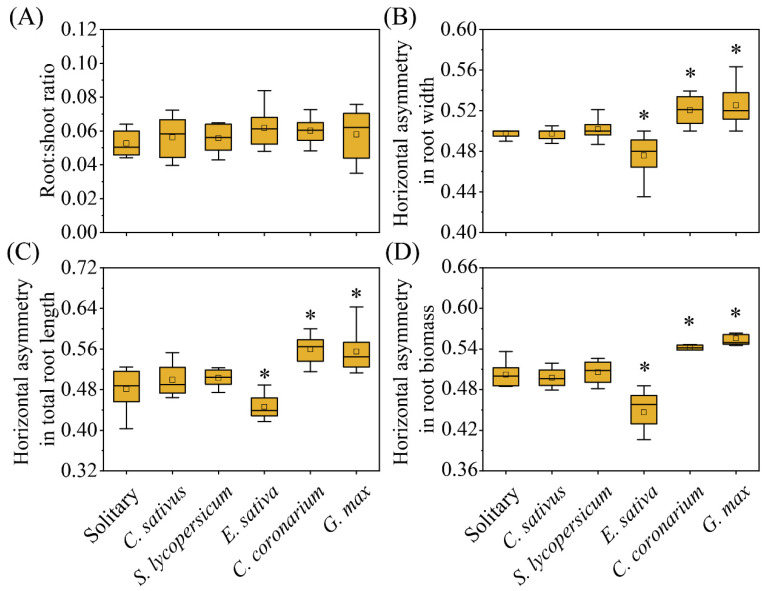
The allocation responses of *C. sativus* to different competition treatments. * indicates a significant difference in (**A**) root:shoot radio, (**B**) horizontal asymmetry in root width, (**C**) horizontal asymmetry in total root length, and (**D**) horizontal asymmetry in root biomass between competition and solitary treatment (*p* < 0.05).

**Figure 4 plants-12-00139-f004:**
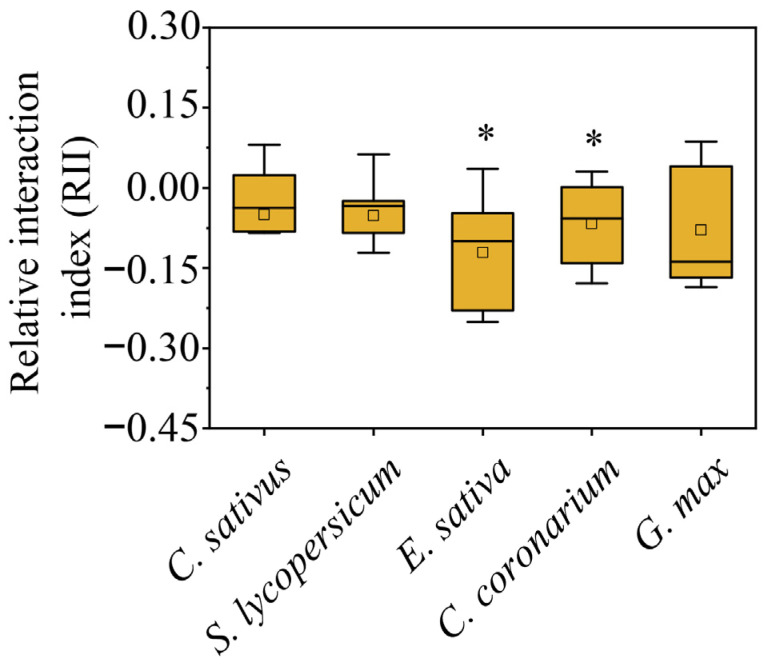
Relative interaction index (RII) for focal plant biomass. * indicates a significant difference between competition and solitary treatment (*p* < 0.05).

**Figure 5 plants-12-00139-f005:**
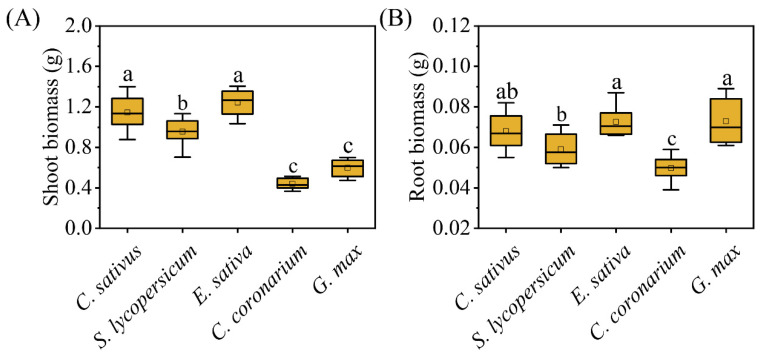
Shoot and root biomass of neighbors in the competition treatments. Letters indicate a significant difference between different neighbors (*p* < 0.05). (**A**) shoot and (**B**) root biomass.

**Figure 6 plants-12-00139-f006:**
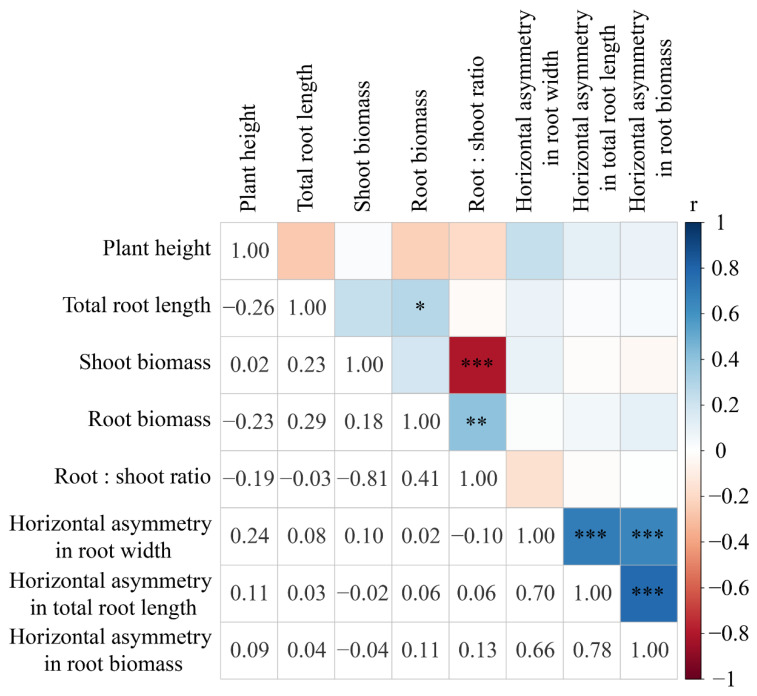
Correlation analysis between 8 response variables. * indicates a significant difference (*p* < 0.05), ** indicates a significant difference (*p* < 0.01), and *** indicates a significant difference (*p* < 0.001).

**Figure 7 plants-12-00139-f007:**
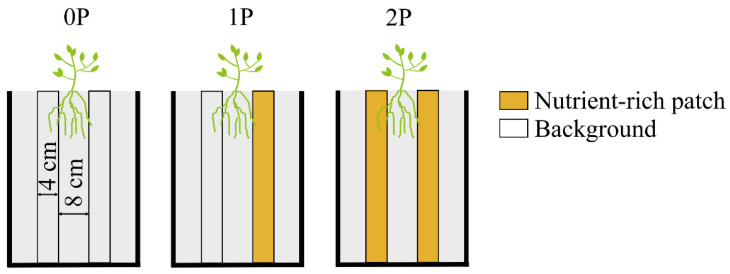
Schematic of heterogeneous nutrient treatments.

**Figure 8 plants-12-00139-f008:**
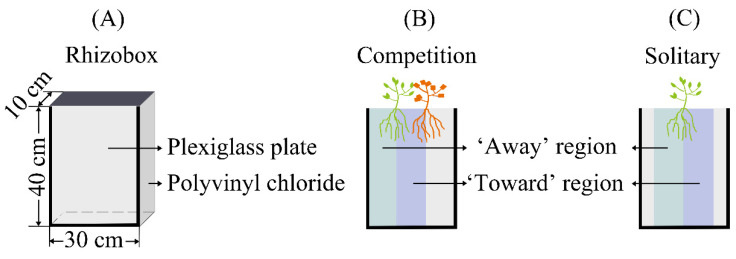
Illustration of the experimental rhizobox and design. (**A**) The structure and size of the rhizobox; (**B**) neighbor treatment (competition); (**C**) no neighbor treatment (solitary). Roots grow away from the neighbor in the ‘away’ region and those towards the neighbor in the ‘towards’ region.

**Table 1 plants-12-00139-t001:** Definite description of different treatments.

Treatments	Description
Solitary	A *C. sativus* plant
*C. sativus*	A pair of *C. sativus* plants
*S. lycopersicum*	A *C. sativus* plant and a *S. lycopersicum* neighbor
*E. sativa*	A *C. sativus* plant and a *E. sativa* neighbor
*C. coronarium*	A *C. sativus* plant and a *C. coronarium* neighbor
*G. max*	A *C. sativus* plant and a *G. max* neighbor

**Table 2 plants-12-00139-t002:** Definite description of 8 response variables.

Response Variables	Description
Plant height	The height (cm) of the aboveground part of the plant (cm)
Total root length (cm)	Total length of roots analyzed using LA-S (Wseen, Hangzhou, China)
Shoot biomass	Dry weight (g) of the aboveground part of the plant (g)
Root biomass	Dry weight (g) of the belowground part of the plant (g)
Root:shoot ratio	Ratio of root biomass to shoot biomass
Horizontal asymmetry in root width	Proportion of the root width for the ‘Toward’ region of focal species
Horizontal asymmetry in total root length	Proportion of the total root length for the ‘Toward’ region of focal species
Horizontal asymmetry in root biomass	Proportion of the root biomass for the ‘Toward’ region of focal species

## Data Availability

Not applicable.

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
