# Peer review of "Response of Cucumis sativus to Neighbors in a Species-Specific Manner"

_plants, 2022, doi:10.3390/plants12010139_

Round 1

Reviewer 1 Report

Dear authors,

Your manuscript is focused on one of the popular plant species in agriculture and its biotic/allelopathic relationships, so it can have a significant practical importance.

I would like to recommend some corrections of the text proposed as follows:

1) Lines 106, 110, 201 - Please, turn the Latin name of species in italic

2) The Conclusion section is too short.

As a whole, your study is promising, but still needs a further development. At this stage, your findings could only be used as a scientific basis for future experiment.

Reviewer 2 Report

Dear Authors

The first sentence of the Abstract is too common. Can you make it more precise and related to your research work?

Maybe you can have a table to clarify the treatment in the material and method section. 

In my opinion,  3 pictures in the supplementary can be inside the manuscript. its more understandable for the target reader. (specially S1).

Reviewer 3 Report

In the article 'Cucumis sativus plants response to neighbors in a species specific manner' authors decribed behavioral outcomes in the growth and allocation responses to 5 different species belonging to 5 families in C. sativus to better understand behavioral strategies of C. sativus plants in response to different neighbors. I believe that the results presented are worthy of publication. I have some questions/comments perhaps to be discussed in the discussion section.

1. Have you considered treating cucumber seedlings with extracts from these plants to see the effect of these extracts? Perhaps such studies for cucumber already exist. Their addition would enhance discussions.

2. Have you noticed other "competition symptoms" on cucumber seedlings, for example: brighter leaves, weak shoots, plant malnutrition, etc.? It would be good to add to the supplementary files sample photos of the experiment showing the whole plants (the aerial part of plants and roots).
